# A Microbiological Approach to Alleviate Soil Replant Syndrome in Peaches

**DOI:** 10.3390/microorganisms11061448

**Published:** 2023-05-30

**Authors:** Derek R. Newberger, Ioannis S. Minas, Daniel K. Manter, Jorge M. Vivanco

**Affiliations:** 1Department of Horticulture and Landscape Architecture, Colorado State University, Fort Collins, CO 80523, USA; derek.newberger@colostate.edu (D.R.N.); ioannis.minas@colostate.edu (I.S.M.); 2Agricultural Research Service, United States Department of Agriculture, Fort Collins, CO 80526, USA; daniel.manter@usda.gov

**Keywords:** plant performance, replant syndrome, rhizosphere, soil bacteriome

## Abstract

Replant syndrome (RS) is a global problem characterized by reduced growth, production life, and yields of tree fruit/nut orchards. RS etiology is unclear, but repeated monoculture plantings are thought to develop a pathogenic soil microbiome. This study aimed to evaluate a biological approach that could reduce RS in peach (*Prunus persica*) orchards by developing a healthy soil bacteriome. Soil disinfection via autoclave followed by cover cropping and cover crop incorporation was found to distinctly alter the peach soil bacteriome but did not affect the RS etiology of RS-susceptible ‘Lovell’ peach seedlings. In contrast, non-autoclaved soil followed by cover cropping and incorporation altered the soil bacteriome to a lesser degree than autoclaving but induced significant peach growth. Non-autoclaved and autoclaved soil bacteriomes were compared to highlight bacterial taxa promoted by soil disinfection prior to growing peaches. Differential abundance shows a loss of potentially beneficial bacteria due to soil disinfection. The treatment with the highest peach biomass was non-autoclaved soil with a cover crop history of alfalfa, corn, and tomato. Beneficial bacterial species that were cultivated exclusively in the peach rhizosphere of non-autoclaved soils with a cover crop history were *Paenibacillus castaneae* and *Bellilinea caldifistulae*. In summary, the non-autoclaved soils show continuous enhancement of beneficial bacteria at each cropping phase, culminating in an enriched rhizosphere which may help alleviate RS in peaches.

## 1. Introduction

Replant syndrome (RS), commonly referred to as replant disease, is a global soil-related challenge induced in trees newly planted upon old orchard sites where repeated monoculture leads to stunted tree growth and reduced yields [1,2,3,4]. RS etiology is not fully understood, but reduced orchard productivity due to RS is caused by a microbial complex of phytopathogens/competitors [5]. Replant symptoms are nonspecific, affect multiple genera of fruit trees, and often correlate with pathogenic generalists such as root-lesion nematodes and *Fusarium* spp. [6,7,8]. Biotic factors such as microorganisms contribute to RS, which is supported by studies where *Prunus persica* (peach tree) biomass is higher in autoclaved soils than in non-autoclaved soils [9]. Nonetheless, abiotic factors such as decreased soil fertility, poor soil structure, and nonoptimal pH can exacerbate RS [5,10]. The consensus that the previous plant of a similar genotype is responsible for initiating RS has supported the notion of intraspecific allelopathy, known as autotoxicity, is a contributing factor [10,11]. However, RS can persist in soils for several years, and it is unknown whether these chemicals are stable for years [9]. Recent understanding of RS and autotoxicity suggests these chemicals are rapidly degraded by rhizosphere and soil microbes but may induce a microbial composition shift in the soil from beneficials to pathogenic or nutrient competitors [10].

The soil microbiome is highly connected, and disturbances can affect bacteriome composition and functionality [12]. Cover crops, tillage, solarization, and fumigation can change microbial communities. Previous management of RS involved chemical fumigation of orchard soils before planting seedlings [4,13]. Soil fumigation induced a growth response for trees in RS soils which lasted one year, but RS symptoms reappeared within two growing seasons [14]. Environmental regulations restrict the use of fumigants that restrict their use in orchards [4]. Thus, sustainable soil practices are needed to alleviate RS.

Cover cropping is a sustainable soil strategy where crops are planted to regenerate soil health rather than to be harvested for economic value. Cover cropping can conserve soil, decrease water runoff, and enhance soil organic matter content [15]. Since cover crops affect the chemical and physical properties of the soil, they also modify the biological properties of the rhizosphere, i.e., the narrow region of soil where root–microbial associations occur [16]. Root–microbial associations within the rhizosphere can potentially improve soil fertility and degrade toxic chemicals [17]. Furthermore, beneficial associations in the rhizosphere can influence pathogen populations [18].

In the current study, alfalfa, fescue, corn, and tomato were tested as cover crops for the purpose of reducing RS symptoms in peach. Additionally, autoclaving was used to determine if the benefits of soil disinfection could complement those of cover crops. Previous studies focused on the identification of reoccurring phytopathogenic instigators of RS, such as fungi, oomycetes, and nematodes. The scope of this study was to emphasize sustainable agricultural techniques that promoted peach health and to identify potential plant growth promoting rhizobacteria for future inoculation studies. These findings show some drawbacks of soil disinfection with cover cropping as a favorable soil regenerative strategy. Furthermore, correlations between microbial taxa and RS alleviation in peaches were identified.

## 2. Materials and Methods

### 2.1. Soil Sampling Site and Disinfection

RS soil for the experiment was acquired from a peach orchard research block, which was established in 2007 at the Colorado State University’s experimental orchard at the Western Colorado Research Center in Orchard Mesa, CO. This peach orchard was established using *Prunus persica* (peach) ‘Cresthaven’ scions with grafted peach ‘Lovell’ rootstocks. The soils from this area have been described as Billings silty clay loam (calcareous, mesic Typic Torrifluvents). RS soils were transported to Colorado State University’s Horticultural Center.

In the Horticultural Center, soils were passed through a metal sieve (2 cm wide) and homogenized. Samples of the replant soil were collected before and after autoclaving and stored at −80 °C to be used as controls for soil bacteriome analysis. Soil was placed in autoclave bags and then in a STERIS brand steam autoclave set on the 40 min liquid cycle at 121 °C, which was run three times. In between cycles, bags carrying the soil were shaken to redistribute the soil before being returned to the autoclave for a second and third time.

Then, 4 L black plastic pots (n = 100) were lined with Vigoro weed control fabric medium duty, placed on Vigoro 15.24 cm plastic plant saucers, and filled with c. 2.1 kg of either untreated RS soil (n = 50) or autoclaved RS soil (n = 50).

### 2.2. Seed Sterilization and Density for Cover Crops

Four crops were selected for this study: natural sweet F1 OG hybrid bicolor Sh2 corn (*Zea mays*), hybrid cherry tomato SUN gold F1 (*Solanum lycopersicum*), ranger alfalfa (*Medicago sativa*), and a fine fescue species mixture of chewing fescue (*Festuca rubra* ssp. *Commutate*), hard fescue (*Festuca longifolia*), and creeping red fescue (*Festuca rubra*). Alfalfa and fescue were selected since these cover crops are known to successfully establish in Colorado soils [19,20]. Tomato and corn have been shown to induce microbial shifts in RS soils under autoclaved conditions, which could potentially create a beneficial microbiome for the incoming peach crop cycle [9]. All cover crops were considered genetically distant from peaches. Seed densities were calculated by using recommended crop seed count or weight per square meter and adjusted by the 0.0222 m^2^ surface area of a 4 L pot. Corn and tomato treatments had one plant per pot [21,22]. For fescue, the recommended use of 50 lbs of seed per acre for high elevation soil in the western United States was used to calculate 0.54 g of fescue seeds per pot [23]. For alfalfa, the recommended use of 75 seeds of alfalfa per square foot was used to calculate 0.038 g of alfalfa seeds per pot [24]. For seed sterilization, 15 mL falcon tubes with seeds were filled with 3% NaOCl and vortexed at max speed (setting 10: 600–2700 RPM) for one minute. NaOCl was removed and seeds were then rinsed with autoclaved distilled water and vortexed at max speed for one minute, with this rinse step being repeated 5 times. Seeds were immediately planted into the soil. Each crop treatment had 10 replicates for autoclaved and non-autoclaved soils. Pots with only autoclaved and non-autoclaved soils served as a no-plant control and were watered to water-holding capacity daily.

### 2.3. Establishing the Cover Crops in a Greenhouse

The experiment was factorial with two factors: soil disinfection (2 levels: autoclave and non-autoclaved) and cover crops (5 levels: corn, tomato, alfalfa, fescue, and a no-plant control). Pots (n = 100) were set in a completely randomized design (5 × 20) using an online random block design generator (https://www.randomizer.org accessed on 26 February 2021) with one treatment per row. Pots were watered at water-holding capacity (c. 200 mL) for six days per week for 12 weeks. After 12 weeks, bulk soil samples were collected with a hand-sized soil probe either in the center of the pot or 2 cm from the base of the plant at a depth of 7 cm. Above ground crop biomass was cut into <2 cm pieces using scissors that were washed in 3% NaOCl followed by heat sterilization using a Bacti-Cinerator III™ from Monoject (St. Louis, MO, USA) in between samples. Above ground fresh biomass of cover crops was recorded, immediately incorporated into the soil within the first 3 cm of the same pot in which the crops had been grown, and left to decompose. After two weeks, bulk soil samples were collected.

### 2.4. Continued Greenhouse Experiment with Peaches

‘Lovell’ (*Prunus persica*) rootstock cultivar was grown from seeds in liners using pro-mix potting media in a greenhouse for 28 days. This RS-susceptible ‘Lovell’ was selected since this rootstock cultivar was grown in the orchard where the RS soil was collected. These four-week-old peach seedlings were transplanted into the pots that previously had cover crops and no-cover-crop controls. Peach seedlings were watered daily with c. 150 mL of tap water. Weeds and cover crops were continuously removed, and no fertilizer was added. Peaches grew for 22 weeks. For microbial analysis, bulk soil and rhizosphere soil was collected. Using a soil probe, bulk soil samples were collected from the top 7 cm of soil within 2 cm of the base of the tree trunk and immediately stored at −20 °C. Rhizosphere was defined as the soil adhering to the roots after the removal of bulk soil and gently shaking the root system. Rhizosphere soil was taken from light colored roots, placed into 15 mL falcon tubes, and immediately stored at −20 °C. Remaining soil on root systems were removed with tap water. Biomass was separated as either above- or below-ground and its weight was recorded. Fresh biomass samples were oven-dried at 90 °C for 72 h and were weighed for above- and below-ground dry biomass. Greenhouse experiments ran between 20 June and 1 August, humidity set point was 70%, cool set point was 24–26.5 °C, heat set point was 18–21 °C, relative humidity ranged from 21–80% (average = 55.5%), and actual temperature ranged from 18.9–38.3 °C (average = 25.4 °C).

### 2.5. Soil Analysis

Soil analysis (total nutrient digest and Haney H_2_O extract) was performed by WARD Laboratories, Inc. (Kearney, NE, USA) [25] on three bulk soil samples per treatment. Total nutrient digest analysis quantified the total values of elements in a soil (C, N, P, K, Ca, S, Mg, B, Zn, Mn, Fe, Cu, Mo). The Haney test uses different extracts from traditional soil test labs, and the extract analysis quantifies nutrients within the soil that are available to soil microorganisms by measuring soil respiration, water-soluble organic carbon, and nitrogen. Soil analyses of total nutrient digest and Haney H_2_O extract by soil treatment of autoclave, cover crop, and control treatments can be found in Appendix A.

### 2.6. DNA Extraction

Total genomic DNA (gDNA) was extracted from 0.25 g of bulk and peach rhizosphere soil in a QIAcube instrument (Qiagen, Germantown, MD, USA) using PowerSoil^®^ DNA kits by Qiagen. All DNA extractions were performed according to Qiagen’s instructions with a final elution volume of 100 μL. DNA concentration was quantified using a Qubit with broad range assay solutions. Of the ten replicates used for biomass, a subset of five replicates were used for bacterial DNA microbial analysis. Bulk soil samples were taken after cover crops were grown for 12 weeks, after cover crops had been incorporated for two weeks, and after peach trees had been growing for 22 weeks. The controls used were pre-extracted Zymo gDNA (Zymo Research Corporation, Irvine, CA, USA) (n = 4), HPLC water (n = 3), stock soil (n = 4), non-autoclaved soil (n = 5), and autoclaved soil (n = 4). In total, 220 samples were extracted.

### 2.7. Oxford Nanopore Library Preparation, Sequencing, and Bioinformatics Pipeline

Based on Qubit concentrations (ng/μL), extracted DNA was diluted 10× with HPLC water to lower DNA concentrations and minimize potential PCR inhibitors. Mastermix consisted of 10 μL Phusion HSII master mix, 7.2 μL H_2_O, 0.4 μL forward primer, and 0.4 μL reverse primer for a total of 18 μL Mastermix per 2 μL sample. Bacterial primers used were Bact_27F-Mn (5′-TTTCTGTTGGTGCTGATATTGCAGRGTTYGATYMTGGCTCAG-3′) and Bact_1492R-Mn (5′-ACTTGCCTGTCGCTCTATCTTC TACCTTGTTACGACTT-3′). Polymerase chain reaction (PCR) settings were 98 °C for 30 s, 98 °C for 15 s, 50 °C for 15 s, and 72 °C for 60 s for 25 cycles, and 72 °C for 5 min. After PCR, equal volumes of DNA and beads were mixed. A 96-pronged magnetic stand moved beads with adhering DNA into two 30 s rinses of 70% EtOH. DNA was eluted in a 96-well plate with 40 µL PCR grade water and beads were removed using a magnetic stand. DNA was quantified using a Qubit with high sensitivity assay solutions. The second PCR settings were 98 °C for 30 s, 98 °C for 15 s, 62 °C for 15 s, and 72 °C for 60 s for 25 cycles, and 72 °C for 5 min.

After a second PCR, DNA and barcodes were pooled in AMPure bead solution in a 96-well plate. Wells with suspended DNA and barcodes were pooled into a clean Lo-Bind tube. MinION sequencer was loaded with a flow cell (R9.4.1) and was prepared for DNA loading. To prepare the flow cell, air (c. 20 µL) was removed using a pipette. The flow cell was then primed with flush buffer, and pooled DNA was loaded into the sampling port. MinKNOW software (v23.04.5) was used to sequence the pooled library for 48 h. Raw data were downloaded and converted into fastq file format using Guppy_basecaller (v6.0.1). Barcodes were sorted by de-multiplex using Guppy_barcoder using barcode kit EXP-PBC096, trimmed, and reads were then filtered by quality and length (Filtlong minimum length: 1000 and mean quality: 70) (Cutadapt: -m 1000 -M 2000). Chimeras were identified and removed by Vsearch. Bacterial taxa were identified using EMU NCBI Reference Database. EMU error correction removed identified bacterial taxa based on alignment and abundance profiles. Bacterial taxa with <one per 10,000 reads were removed. Sequencing data came from three separate sequence runs, which were pooled for data analysis.

### 2.8. Statistical Analysis

All statistical analyses were performed in RStudio Version 1.4.1103. A non-parametric test, the Kruskal–Wallis rank sum test, was used to analyze fresh cover crop biomass and peach dry biomass by soil treatment (autoclaved vs. non-autoclaved). Pairwise comparisons using the Wilcoxon rank sum exact test was used to infer differences between plant biomass and soil treatment. For regressions analyzing soil nutrients from the end of the peach experiment, peach dry biomass was used. The Lagrange multiplier test was used for the regressions fit with broom and tidyverse packages in RStudio. The vegan package was used to test for significant differences between treatments with perMANOVA and visualized with a constrained principal coordinate analysis (PCoA). Bray–Cutis was used to determine distance for PCoAs. Homogeneity of multivariate dispersions was measured using betadisper from the vegan package. Differential abundance analysis was based on bacterial species counts that were transformed using a log2 fold-change and the Benjamini and Hochberg statistical method using the false discovery rate function (FDR) and 0.05 as the accepted threshold for the adjusted *p*-value.

## 3. Results

### 3.1. Effect of Soil Disinfection on Cover Crop Biomass

The Kruskal–Wallis test for cover crop above-ground biomass shows the soil treatment (autoclaved vs. non-autoclaved) is significant, χ^2^ = 16.398 (df = 1, *p*-value < 0.001). All cover crops grown in autoclaved soils have a higher biomass than cover crops grown in non-autoclaved soils. Biomass of corn (*p* < 0.001), fescue (*p* < 0.001), and tomato (*p* < 0.001) are significantly different between their respective autoclaved and non-autoclaved treatments. However, alfalfa crop biomass is not significantly different (*p* = 0.459) between the autoclaved and non-autoclaved soil treatments (Figure 1). Alfalfa’s biomass in non-autoclaved soils shows a 34.6% reduction compared to alfalfa grown in autoclaved soils. Corn grown in autoclaved soils has the highest biomass out of all autoclaved and non-autoclaved cover crop treatments. In this study, tomato plants grown in untreated soils (RS soils) have a biomass reduction of 85.8% compared to tomato plants grown in autoclaved soils. This supports the trend that soil disinfection improves plant health.

### 3.2. Effect of Cover Crop and Biomass Incorporation on the Soil Microbiome

The bulk soil bacteriome where cover crops had been growing for 12 weeks were analyzed. The perMANOVA test shows that crop type (*p* = 0.001), autoclaved and non-autoclaved (*p* = 0.001), and the interaction (*p* = 0.03) between the two factors are significant and the CAP axes explain a total of 37.9% of the variance for all samples (Figure 2A). Separation between autoclaved and non-autoclaved soils is clear along axis 1 and explains 29.9% of the variance. For cover crops, autoclaved soils (average distance to median: 0.400) have a lower dispersion than non-autoclaved soils (average distance to median: 0.411) (Figure 2A). Within the autoclaved soil treatment, cover crop bulk soil microbiomes overlap while cover crop treatment has a greater role in shaping the microbiome in non-autoclaved soils. Corn grown in autoclaved soils has the highest biomass after 12 weeks of growth, but its microbiome does not show clear separation from the other cover crops. In the bulk soil of the cover crops, no-cover-crop controls overlap with the crop treatments in either of their respective soil treatments.

Microbes from the autoclaved soil treatment of cover crop bulk soils were of interest due to the increase in biomass in all cover crops. Although the positive effects of autoclaving on cover crop growth are primarily due to the removal of or reduction in potentially negative microorganisms, this study aimed to identify beneficial bacteria instead of highlighting deleterious bacteria that has been previously studied. Thus, differential abundance between non-autoclaved and autoclaved cover crop bulk soils highlights bacteria whose abundances are significantly different (Appendix A). There are 14 bacterial taxa whose abundance is driven by autoclaving, since all cover crops and no-cover-crop controls share these microbes. No common bacteria are found to be promoted within just the four crop treatments (indicated by the orange circle in Figure 2B). Autoclaved treatments with the highest unique bacterial taxa are no crop (n = 48) and fescue (n = 33), followed by tomato (n = 13), corn (n = 7), and alfalfa (n = 3). In addition, non-autoclaved cover crop bulk soil treatments show 26 bacterial taxa whose abundances are higher than in autoclaved soils and are shared within all cover crop treatments (Appendix A).

Bulk soils after the cover crop had been incorporated and decomposed for two weeks continued to show separation between autoclaved and non-autoclaved microbiomes (Figure 3). The perMANOVA test shows that cover crop history (*p* = 0.001) and autoclaved soil treatment (*p* = 0.001) are significant and explain a total of 35.7% of the variance. The interaction between cover crop history and autoclaved soil treatment is not significant (*p* = 0.105) (Figure 3). Similar as in crop history, the microbiome corresponding to bulk soil after cover crop incorporation shows a tighter cluster in autoclaved soils (Figure 3). Interestingly, the incorporation of alfalfa biomass in non-autoclaved soils shows an independent cluster compared to other cover crop treatments. Bacterial drivers (identified by the differential abundance) of the incorporated cover crop bulk soil microbiome that are found in all crops and no-cover-crop controls continue to have increased abundance in their respective autoclaved soil treatment (Figure 3). In autoclaved cover-crop-incorporated soils, the bacterial species *Tumebacillus soli*, *Cytobacillus oceanisediminis*, and *Mesobacillus subterraneus* are found to be bacterial drivers of the microbiome (Figure 3) and these are the same microbes found in autoclaved cover crop soils (Appendix A). In non-autoclaved soil incorporated with cover crops, the bacterial species *Vicinamibacter silvestris*, *Skermanella stibiiresistens*, *Bacillus megaterium*, *Nostoc* sp. HK-01, and *Nostoc* sp. PCC 7107 are primarily found (Figure 3), which are also present in non-autoclaved cover crop soils (Appendix A).

### 3.3. Effect of Soil Disinfection and Cover Crop Incorporation on Peach Growth

The Kruskal–Wallis test for peach dry total biomass shows that the soil treatment effect (autoclaved vs. non-autoclaved) is significant (χ^2^ = 35.298, df = 1, *p*-value < 0.001). Biomass is higher for peach trees grown in soil that has not been disinfected via steam autoclave, and it is observed for most soil cover crop treatments, with alfalfa being the exception (*p* = 0.095) (Figure 4A). Pairwise comparisons using corn (*p* = 0.002), fescue (*p* < 0.001), and tomato (*p* < 0.001) cover crops show a significant difference in biomass within autoclaved and non-autoclaved soil treatment pairs (Figure 4A). Between the two no crop controls that later had peaches growing, there was no significant difference in biomass within autoclaved and non-autoclaved soil treatments (*p* = 0.363) (Figure 4B). Additionally, autoclaved soils with a cover crop history of fescue (*p* < 0.001) and corn (*p* = 0.002) perform worse than autoclaved soils with a history of no cover crops (Figure 4B). Within autoclaved soil treatments, peaches grown in alfalfa and tomato have a higher biomass than peach trees in soils that previously had corn and fescue. In all, peach trees grown in non-autoclaved soils have the highest biomass compared to peaches grown in autoclaved soils.

### 3.4. Nitrogen and Nutrient Analysis

Nutrient analyses were performed to investigate if nutrient cycling could help explain the differences between autoclaved and non-autoclaved soil treatments. Dry peach biomass was used in regression plots with several different soil nutrient parameters. Soil nutrient parameters that are not significant predictors of peach biomass are total organic carbon, organic nitrogen, organic C/N ratio, total phosphorus (H3A), available phosphorus, potassium (H3A), and available potassium. The only positive correlation between dry peach biomass is with available organic nitrogen (R^2^ = 0.144, *p*-value = 0.038) (Figure 5). Available nitrogen is statistically different by crop treatment with alfalfa and tomato having higher available nitrogen than fescue and no cover crop treatments (Appendix A). Overall, alfalfa and tomato treatments have the highest available nitrogen and are not statistically different compared to corn treatments. Fescue and no crop treatments have lower available nitrogen (Figure 5). The only negative correlation between dry peach biomass found to be significant is with ammonium (Appendix A).

### 3.5. Effect of Soil Disinfection, Cover Crop Incorporation, and Peach Growth on the Bulk and Rhizosphere Soil Bacteriome

Shannon index of controls and all treatments separated by soil and cover crop history shows a trend of non-autoclaved soils having a greater alpha diversity than autoclaved soils (Appendix A). For beta diversity, the autoclave soil treatment is the driver of cluster separation. Non-autoclaved and autoclaved soil bacteriomes remain separated for the entire study (cover crop bulk soil, cover crop incorporation bulk soil, peach bulk soil, and peach rhizosphere).

The bacteriome corresponding to bulk soil of peach grown under autoclaved (average distance to median: 0.402) and non-autoclaved (average distance to median: 0.387) conditions show that dispersion is greater in disinfested soils. The perMANOVA test shows that cover crop history (*p* = 0.001) and autoclaved soil treatment (*p* = 0.001) is significant and the CAP axes explain a total of 32.5% of the variance. The interaction between cover crop history and autoclaved soil treatment is not significant (*p* = 0.369) (Figure 6A). This result indicates how the autoclaved bacteriome continues to be prone to change, and the separation along axis 2 supports clustering by cover crop history (Figure 6A). Under both soil treatments, it is observed that previous cover crop histories create bacterial associations. Peaches grown in non-autoclaved soils with a cover history of alfalfa create a unique bacteriome and have the highest shift from the non-autoclaved centroid than other non-disinfested treatments (average distance to median for: alfalfa, 0.356; fescue, 0.343; tomato, 0.340; no cover crop, 0.327; corn, 0.279).

Peach seedlings with the highest biomass correspond to non-autoclaved soil treatments. Therefore, microbes from non-autoclaved peach bulk soils are of the most interest. Within all non-autoclaved treatments (cover crop history and the no-cover-crop history control) for the peach crop cycle, there are seven bacterial species (*Bacillus megaterium*, *Brevitalea aridisoli*, *Brevitalea deliciosa*, *Gaiella occulta*, *Nitrospira japonica*, *Skermanella rosea*, *Skermanella stibiiresistens*) whose abundance increases compared to autoclaved peach bulk soils (Appendix A). Soils with cover crop histories have no additional bacterial species in common that do not increase in the no cover crop treatment (indicated by the orange circle; (Figure 6B). Non-autoclaved treatments with the highest unique bacterial species that increase compared to autoclaved soil correspond to no crop and fescue.

In contrast to the peach bulk soil, the rhizosphere soil corresponding to non-autoclaved treatment shows a tighter cluster than the rhizosphere soil of autoclaved soils (Figure 7A). Peaches grown in autoclaved soils (average distance to median: 0.319) loosely cluster based on previous cover crop and show overlap. Within non-autoclaved soils (average distance to median: 0.36), peaches that previously had a cover crop of alfalfa or fescue samples have the greatest shift away from the no crop control. The constrained PCoA shows that cover crop history (*p* = 0.001) and autoclaved soil treatment (*p* = 0.001) are significant and explain a total of 50.6% of the variance. The interaction between crop history and autoclaved soil treatment is not significant (*p* = 0.267) (Figure 7A). Similar to the peach bulk soil, the soil history of alfalfa grown in non-autoclaved soils develops a distinct bacteriome.

From the differential abundance of non-autoclaved peach bulk soil, out of the seven microbes found in all treatments, six of these bacterial species (*Bacillus megaterium*, *Brevitalea aridisoli*, *Brevitalea delicios*, *Gaiella occulta*, *Nitrospira japonica*, *Skermanella rosea*) are found again in all non-autoclaved peach rhizosphere soil treatments, with *Skermanella stibiiresistens* being the exception (Appendix A). Differential abundance of non-autoclaved peach rhizosphere soil per crop shows that there are 11 shared bacterial species that increase in abundance by soil treatment, regardless of cover crop (Figure 7B). Soils with cover crop histories have two additional bacteria species, *Paenibacillus castaneae* and *Bellilinea caldifistulae,* in common (Figure 7B). Bacterial species that are in higher abundancies in non-autoclaved peach rhizosphere soils with a cover crop treatment and not found in the RS symptomatic non-autoclaved soils without a crop control are *Bacillus cereus* (alfalfa, fescue, and corn), *Paenibacillus xylanilyticus* (fescue, corn, and tomato), *Baekduia soli* (corn and tomato), *Terrimonas suqianensis* (corn and tomato), *Desulfobulbus propionicus* (fescue and corn), *Paenisporosarcina indica* (alfalfa and corn), *Desulfopila inferna* (alfalfa and fescue), and *Desulfotalea psychrophile* (alfalfa and fescue) (Appendix A). Non-autoclaved no-crop treatments, notably, have the highest unique bacterial taxa (n = 53). Of the cover crops, corn (n = 10) and fescue (n = 10) have the highest counts of unique bacterial taxa in higher abundances, with tomato (n = 8) and alfalfa (n = 5) having the least (Figure 7B).

## 4. Discussion

### 4.1. Cover Crop Biomass and Bulk Soil Bacteriome

Biomass for all cover crop treatments is significantly higher in autoclaved soils. Pathogen accumulation in soils has been observed in continuous monocultures of a wide range of crops [11]. Steam autoclave as a soil disinfection method has been shown to increase crop biomass by having decreased bacterial populations, particularly those that could have competed for nutrients and are pathogenic to the plant [9,26,27,28]. Although the mycobiome was not assessed in this study, conceptually, disinfestation would also reduce deleterious or pathogenic fungi. This coincides with the premise that non-native plants can acquire higher yields due to the low amount of specialized soil microbial pathogens and nutrient competitors [29]. Tomato’s biomass reduction in non-autoclaved soils suggests that they are less tolerant to RS-associated microbes than alfalfa, whose biomass reduction is less severe. Plant fitness is dependent on the strong associations with soil microbiota, and the biomass results show a strong biological component in RS soils.

Microbes associated with peach RS negatively affect cover crop biomass of genetically distant crops. However, cover crops do not appear to establish their own distinct beneficial bacteriome, as all cover crop and no crop control bacteriomes overlap (Figure 2A). Beneficial bacterial taxa are identified in autoclaved cover crop soils, most of which fall into the categories of antimicrobial, toxic metal bioremediation or uptake, and nitrogen-related activities (Appendix A). Microbes in autoclaved cover crop bulk soils with antimicrobial properties are *Janthinobacterium* sp. strain *Marseille* (antifungal), *Pseudomonas koreensis* (antifungal), *Paenibacillus typhae* (antifungal), *Cytobacillus oceanisediminis* (antibiotic), *Fictibacillus phosphorivorans* (nematicidal), *Fictibacillus arsenicus* (nematicidal), and *Verrucomicrobium spinosum* (nematicidal) [30,31,32,33,34,35,36].

Microbes in autoclaved cover crop bulk soils with toxic metal bioremediation or uptake properties are *Thermincola potens*, *Arthrobacter* sp. PGP41, *Brevundimonas diminuta*, and *Ensifer adhaerens* [37,38,39,40,41]. Microbes in autoclaved cover crop bulk soils with nitrogen-related activities are *Bacillus dakarensis*, *Nostoc punctiforme*, and *Pseudomonas koreensis* [33,42,43] Autoclaving the soil temporarily reduces RS symptoms and maintains beneficial microbes.

### 4.2. Higher Peach Biomass in Non-Autoclaved Soil and Its Bacteriome

Biomass is higher in autoclaved soils for the cover crop portion of the study; however, the peach seedlings show the highest biomass results when grown in soils that have never been autoclaved. Soils of the autoclaved treatment were only autoclaved once, which was immediately before the plantings of the cover crops. Cover crops were incorporated into the same soil in which they had been grown, and peach seedlings were then planted. Non-autoclaved treatments with a cover crop history of alfalfa, corn, and tomato show a higher peach seedling biomass than the non-autoclaved treatments with no history of a cover crop (Figure 4). Autoclaved treatments with a cover crop history do not outperform autoclaved treatments with no-cover-crop history in terms of peach seedling biomass. The benefit of autoclaving the soil is lost after the cover crop cycle. In all, peach biomass is cover crop treatment-dependent, and the benefit of soil disinfection increasing crop biomass is short term.

Autoclaving soils show slight changes in nutrient accumulation with some changes in plant macronutrients being inconsistent [9,44,45]. In this study, trends such as a slight increase in nitrate in autoclaved soils and in ammonium in non-autoclaved soils by crop treatment are observed. Available nitrogen is positively correlated with peach biomass. Additionally, available nitrogen is significantly higher in tomato and alfalfa compared to fescue and no cover crop treatments (Appendix A). Field studies show that increasing crop biomass enhances weed suppression, decreases nitrate leaching (improved C/N ratio) and above-ground biomass N. However, a study found an increase in crop biomass negatively impacted inorganic N availability and the following cash crop’s (corn) yield was decreased [46]. Similarly, in this study, a decrease in ammonium is correlated with an increase in peach seedling biomass. Overall, nutrients in the soil do not explain the decrease/increase in peach biomass between the autoclaved and non-autoclaved soils.

In the present study, while autoclaving the soil increases cover crop biomass, these same autoclaved soils do not increase peach biomass as the following crop cycle. This could be due to the non-bacterial RS pathogens recuperating to previous microbial compositions pre-disinfection. The benefits of soil disinfection are temporary, even if the impact of the overall bacteriome persists. For instance, methyl bromide treatments produce yield increases for crop cycles following the disinfection of the soil, but strategies are needed to augment the duration of these benefits [47,48]. This would indicate that populations of pathogenic and nutrient-competing microbes recover over time to induce replant symptoms anew. In non-autoclaved peach soils with a cover crop history, which show the highest peach biomass, the beneficial bacterial taxa identified have either antimicrobial, iron-reducing, or nitrogen-related capabilities. Bacterial species with antimicrobial capabilities are *Fimbriiglobus ruber* (antifungal), *Peribacillus simplex* (antifungal), *Bacillus altitudinis* (antifungal), *Stigmatella aurantiaca* (antifungal/antibiotic), *Bacillus halotolerans* (antifungal), *Bacillus megaterium* (antibacterial), *Bacillus cereus* (antifungal), *Bacillus pumilus* (antibiotic), *Paenibacillus castaneae* (nematicidal), and *Bacillus pumilus* (nematicidal) [49,50,51,52,53,54,55,56,57,58,59]. Bacterial taxa with iron-reducing capabilities are *Desulfuromonas michiganensis*, *Desulfuromonas soudanensis*, *Pelobacter carbinolicus*, *Geobacter* sp. M2, *Aciditerrimonas ferrireducens*, *Geobacter bemidjiensis*, *Pedomicrobium Americanum*, *Geobacter uraniireducens*, *Geobacter psychrophilus*, *Pseudomonas sagittaria*, *Paenibacillus guangzhouensis*, and *Pseudarthrobacter* sp. NIBRBAC000502772 [60,61,62,63,64,65,66,67]. Bacterial taxa with nitrogen-related capabilities (nitrogen fixer, nitrogen reducer) are *Gaiella occulta*, *Nitrospira japonica*, *Clostridium magnum*, *Candidatus Saccharibacteria*, *Geobacter* sp., *Microvirga ossetica*, *Azospira restricta*, *Nitrosospira multiformis*, *Paenibacillus massiliensis*, *Paenibacillus xylanilyticus*, *Microvirga zambiensis*, *Aromatoleum aromaticum*, and *Bacillus megaterium* [50,59,68,69,70,71,72,73,74,75,76,77,78]. For the non-autoclaved cover crop treatment, alfalfa’s rhizosphere bacteriome is the treatment that overlaps with the no crop control the least (Figure 7A), and the lack of bacteria species in higher abundancies that are common among multiple cover crops of three or more are few, indicating that there are multiple bacteriomes that can alleviate RS with relatively minimal shifts.

To highlight the loss of key beneficial bacterial species that do not recover their abundancies post-autoclaving, *Bacillus megaterium* is used as an example (Appendix A). This well-documented plant-growth-promoting rhizobacteria has been known to use phytohormones (auxins, gibberellins, and cytokinins), penicillin amidase for biocontrol, and could contain the nifH gene for nitrogen-fixing capabilities [50,58,59,79]. Although bacterial abundancies are affected long term in autoclaved soils, RS-causing microbes are re-established while beneficial bacteria, as identified in non-autoclaved soils, fail to recover.

### 4.3. Reduced Peach Biomass Treatments Show Beneficial Bacteria Instead of a Myriad of Phytopathogens

RS symptoms are observed in all peach seedlings, but these symptoms are exacerbated in peach seedlings grown in soils with a history of autoclaving. It is expected that peaches that have the lowest biomass have bacteriomes that overlap with that of the original RS soils, however, this is not the case. Autoclaved soils with a history of fescue and corn are the treatments with the lowest biomass, and their bacteriomes do not overlap with the initial non-autoclaved bulk soils (Appendix A). These distinct bacteriomes that show RS symptoms indicate that the microbial composition of RS soils is not dominated by an identical community of pathogenic and nutrient-competing microbial taxa, but that the abundancies of only a few taxa are required to cause RS symptoms, such as a decrease in biomass. Li et al. (2019) showed that even a slight shift away from the RS bacteriome of peaches reduced RS symptoms, indicating that a large shift away from the replant soil bacteriome was not required to reduce RS symptoms.

Although the control (non-autoclaved soil with no history of cover crops) has the lowest peach biomass out of the non-autoclaved treatments, the control shows the highest count of unique bacterial taxa (n = 53), many of which have previously been associated with beneficial traits. Antimicrobial bacteria are *Bacillus halotolerans* (antifungal) *Bacillus pumilus* (nematicidal), and *Paenibacillus castaneae* (nematicidal) [49,57]. Also, *Microvirga zambiensis* and *Aromatoleum aromaticum* have previously been associated with the nitrogen cycle [72,77]. Additionally, *Azotobacter chroococcum* and *Bacillus halotolerans* have been known to aid in plant nutrition [80]. Multifunction microbes such as *Bacillus pumilus,* which is capable of gibberellins and antibiotics production, and *Peribacillus simplex* (previously *Bacillus simplex*), which has been shown to synthesize auxin and has anti-fungal activity, are also found in non-autoclaved soils with a no-cover-crop history [54,56,81,82,83]. Although these bacteria have previously been shown to be beneficial, further studies are needed to prove their direct influence on RS.

## 5. Conclusions: A Healthy Peach Bacteriome Progression

A robust population of beneficial bacteria are needed to remedy RS soils. Non-autoclaved soil cultivated with alfalfa, corn, and tomato as cover crops developed the best conditions for peaches to withstand RS in this study. This further supports the idea that certain cover crops may be deployed to reduce RS, specifically for peaches. *Paenibacillus castaneae* and *Bellilinea caldifistulae,* which were cultivated exclusively in the rhizosphere of non-autoclaved soils by peaches for only cover crop histories, may be beneficial and further study could shed light on their role as general colonizers that can possibly reduce RS. Non-autoclaved bulk soils and peach rhizospheres also have an increased abundance of *Bacillus megaterium*, *Gaiella occulta*, and *Nitrospira japonica*. However, these bacterial taxa are also present in the non-autoclaved and no-cover-crop control, which did not outperform the non-autoclaved cover crop treatments (alfalfa, corn, and tomato) in terms of biomass. Nonetheless, further research should be conducted to determine the role of these bacteria in alleviating RS, as these bacteria could be specifically recruited by peaches since abundances are present in the peach rhizosphere in all non-autoclaved treatments. Future studies should use mock community inoculations to investigate the robustness of these bacterial species, since bacteriomes function as a consortia and may require one another to reduce RS.

In contrast, soil disinfection instigates the loss of bacterial species with populations unable to recover within the time frame of this study. This gives an insight into the possible consequences of effective soil disinfection techniques. The Shannon index shows how the newly autoclaved RS bulk soil control has a drastically reduced alpha diversity compared the initial untreated RS bulk soil control. However, the Shannon index supports the fact that many bacterial populations are able to recover by the time the cover crops have grown (Appendix A). This is in line with previous studies that saw benefits of reducing microbial load using soil disinfection techniques and immediately planted peach trees [9,84]. Here, it is proposed that moderate soil disinfection should be used to avoid removing beneficial microbes by using temperatures that are high enough to be lethal to poor soil competitors such as phytopathogens, but low enough for beneficial bacteria to recolonize. The present study shows that cover crops can help ameliorate RS symptoms, but not all cover crops provide equal benefit, with soil disinfection benefits being temporary.

## Figures and Tables

**Figure 1 microorganisms-11-01448-f001:**
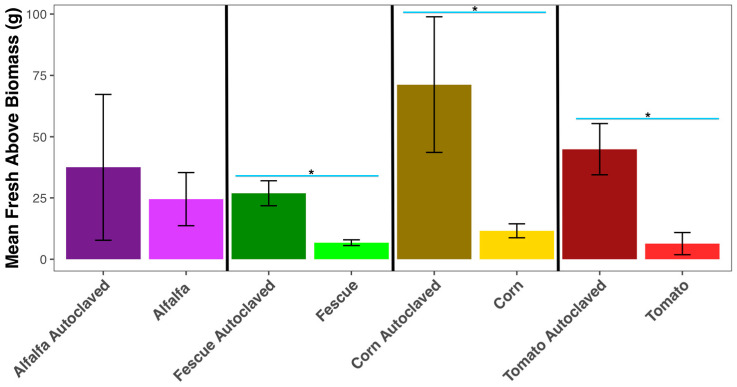
Above-ground fresh biomass of cover crops. Pairwise comparisons using Wilcoxon rank sum exact test to infer significance for each cover crop by autoclaved and non-autoclaved soil treatment. Error bars use the equation ymin/ymax = mean ± standard deviation. Significance (*p* < 0.01) for pairwise comparisons were denoted with *.

**Figure 2 microorganisms-11-01448-f002:**
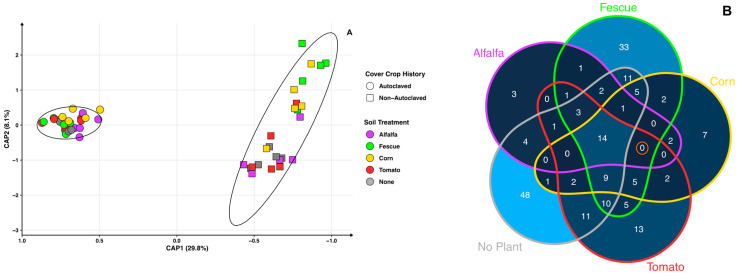
Cover crop bulk soil bacteriomes. (**A**) Constrained principal coordinate analysis (PCoA) using Bray–Curtis distance for cover crop bulk soil bacteriomes. Circle shape represents autoclaved and squares non-autoclaved. Colors indicate cover crop treatment: alfalfa (purple), fescue (green), corn (yellow), tomato (red), and no crop (gray). (**B**) Differential abundance Venn diagram shows a count of bacteria species in higher abundancies in autoclaved compared to non-autoclaved cover crop bulk soils separated by crop. Orange circle indicates common bacteria found to be promoted within all cover crop treatments. Color gradient displays low bacterial taxa species counts in dark blue and higher counts in light blue.

**Figure 3 microorganisms-11-01448-f003:**
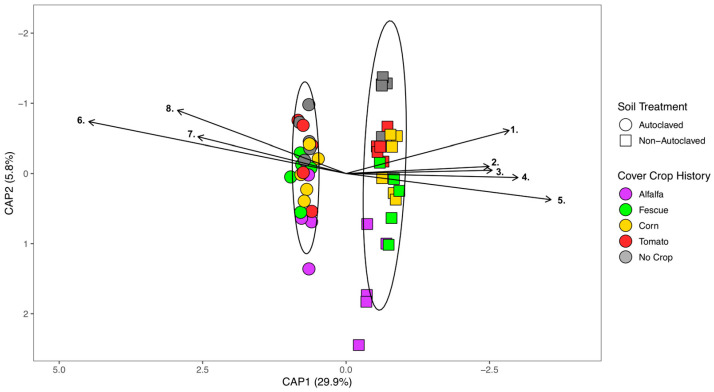
Biplot of a constrained principal coordinate analysis (PCoA) using Bray–Curtis distance for cover-crop-incorporated bulk soil bacteriome. Circles represent autoclaved and squares non-autoclaved. Colors indicate cover crop treatment: alfalfa (purple), fescue (green), corn (yellow), tomato (red), and no crop (gray). Bacterial taxa: 1. *Vicinamibacter silvestris*, 2. *Skermanella stibiiresistens,* 3. *Bacillus megaterium*, 4. *Nostoc* sp. PCC7107, 5. *Nostoc* sp. Hk-01, 6. *Cytobacillus oceanisediminis*, 7. *Mesobacillus subterraneus*, 8. *Tumebacillus soli*.

**Figure 4 microorganisms-11-01448-f004:**
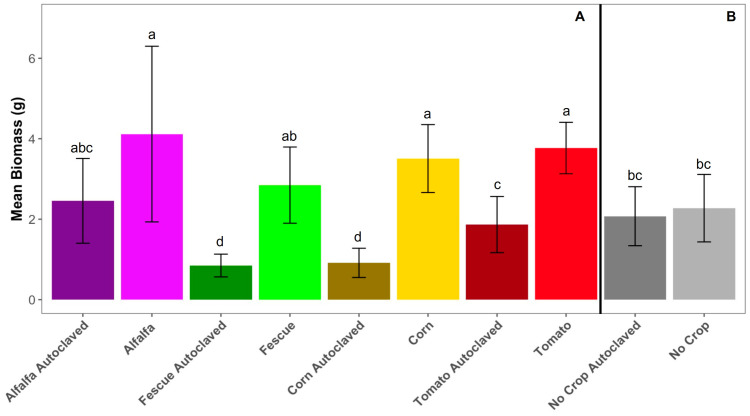
Peach biomass. (**A**) Total dry peach biomass for peach trees with a cover crop history. Wilcoxon rank sum exact test was used to infer significance for each cover crop and autoclaved soil treatment. Dry peach biomass was recorded. Dry biomass was used since it accounted for the fluctuating water concentrations within plant tissues. Error bars use the equation ymin/ymax = mean ± standard deviation. (**B**) Total dry peach biomass for peach trees in autoclaved and non-autoclaved no-cover-crop controls. Both autoclaved and non-autoclaved soils were initially set up at the start of the cover crop experiment, meaning these soils experienced fallowness for 14 weeks before having peach seedlings planted. Error bars use the equation ymin/ymax = mean ± standard deviation. Different superscript letters denote significant difference (*p* < 0.01) compared with different cover crop histories and soil disinfection treatments by Wilcoxon rank sum exact test.

**Figure 5 microorganisms-11-01448-f005:**
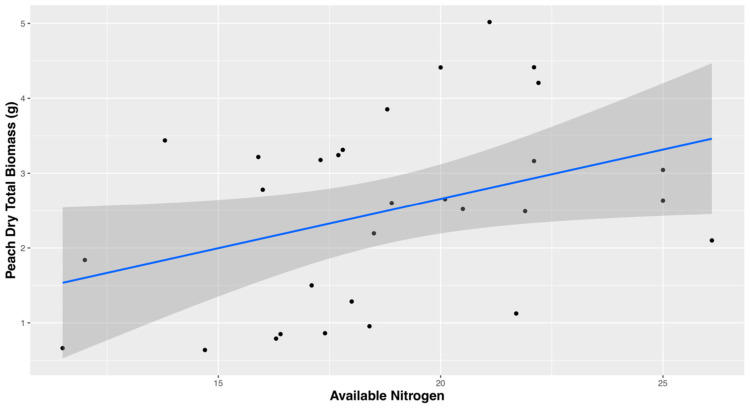
Available organic nitrogen in correlation with total dry peach biomass.

**Figure 6 microorganisms-11-01448-f006:**
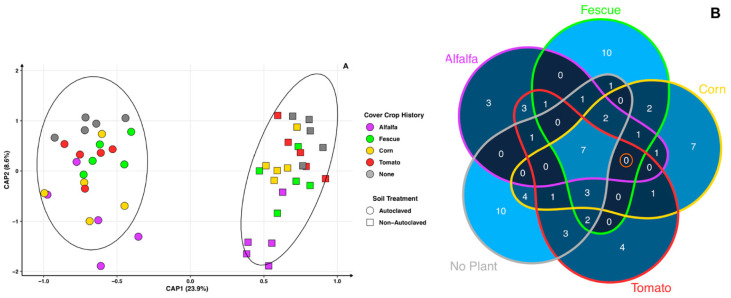
Peach bulk soil bacteriomes. (**A**) Constrained principal coordinate analysis (PCoA) using Bray–Curtis distance for peach bulk soil bacteriomes. Circles represent autoclaved and squares non-autoclaved. Colors indicate cover crop treatment: alfalfa (purple), fescue (green), corn (yellow), tomato (red), and no crop (gray). (**B**) Differential abundance Venn diagram shows a count of bacteria species in higher abundancies in non-autoclaved and autoclaved peach bulk soils separated by crop. Orange circle indicates common bacteria found to be promoted within all cover crop treatments. Color gradient displays low bacterial species counts in dark blue and higher counts in light blue.

**Figure 7 microorganisms-11-01448-f007:**
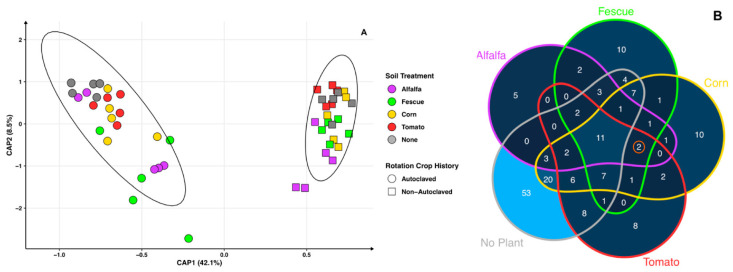
Peach rhizosphere bacteriomes. (**A**) Constrained principal coordinate analysis (PCoA) using Bray–Curtis distance for peach rhizosphere bacteriomes. Circles represent autoclaved and squares non-autoclaved. Colors indicate cover crop treatment: alfalfa (purple), fescue (green), corn (yellow), tomato (red), and no crop (gray). (**B**) Differential abundance Venn diagram shows a count of bacteria species in higher abundancies in non-autoclaved and autoclaved peach rhizosphere soils separated by crop. Orange circle indicates common bacteria found to be promoted within all cover crop treatments. Color gradient displays low bacterial taxa species counts in dark blue and higher counts in light blue.

## Data Availability

DNA sequencing, plant biomass, and soil nutrient datasets generated during and/or analyzed during the current study with R markdown files are on GitHub, [https://github.com/Derek-Newberger/MicroorganismsJournal_2381389_DRN] and [https://github.com/DanielManter-USDA/DRN-2381389]. (Accessed on 23 May 2023). Access to raw data will be granted upon request.

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
