# Peer review of "A Microbiological Approach to Alleviate Soil Replant Syndrome in Peaches"

_microorganisms, 2023, doi:10.3390/microorganisms11061448_

Round 1

Reviewer 1 Report

Manuscript entitled “A microbiological approach to alleviate soil replant syndrome in peaches” written by Newberger et al., explained the replant (RS) on the growth and yield of tree fruit orchards. In addition, author also explained, that by developing/altering health bacteriome we can reduce the RS in peach. The study is well designed, introduction is good, methodology is sound, results are interested having potential of readers interest and manuscript is in the scope of journal. However, I have some suggestions which authors need to be address.

Q1; What is different between soil bacterial microbiome or soil bacteriome? In my opinion the words microbiome is a broad term dealing with all kinds of microorganism including bacteria, fungi, actinomycetes. In this study, authors just only study the soil bacteriome not the soil microbiome, so it’s better to be use a specific word “soil bacteriome” rather than using a broad word “soil bacterial microbiome”.

Q2; What author means by cover crop growth? It is confusing use a proper terminology.

Q3; References are not in a proper order line 32 references start from 45, 50, 75, and 91. Where is reference 1? Same as for line 34, 36, 38, and 39…………………………….

Q4; In the whole manuscript the scientific names are no italic, I do not know why authors are doing like this? I think authors cannot read the manuscript after following the journal format, as the references were also start form 45 not from 1. Authors should adopt a serious behavior when writing and formatting a manuscript.

Line 12; Prunus persica should be italic

Line 13; followed by cover crop growth and incorporation the meanings are not clear need to be revise and same for line 14.

Line 14-16; bacterial soil microbiome needs to be revise in the whole manuscript as “soil bacteriome”

Line 19; Scientific name must be italic, and need to check in the whole manuscript.

Line 36-38; I cannot understand what author want to say. Rewrite the sentence.

Line 47; Can change diversity and composition of ……………

Line 72-75; rewrite the sentence

Line 76; what do you mean by three 40-min liquid cycles?

Line 78; for soil bacteriome analysis.

Line 80-81; each pot was filled with 2.1 kg of soil either untreated ………………………

Line 81-82; delete just below 5 cm of the lip of cup. Each pot contained c. 2.1 kg of soil.

Line 89-92; Reference must be in number format.

Line 101; Everyone knows that how to separate a supernatant or liquid from the falcon tubes.  

The sentence structure is not correct. incomplete senetnces. Authors must revised the manuscript for sentence structure. Authors must delete unnecessary sentences, such as balck plastic, Using a pipette, the NaOCl solution was then aspirated out. 

Author Response

Dear reviewer 1,

Please find our point-by-point response to the comments.

Sincerely,

First Author

Derek R. Newberger, PhD Candidate

Graduate research assistant, Department of Horticulture and Landscape Architecture

Colorado State University

Derek.Newberger@colostate.edu

Dr. Jorge M. Vivanco, PhD

Professor, Department of Horticulture and Landscape Architecture

Colorado State University

J.Vivanco@colostate.edu

Reviewer 1:

Manuscript entitled “A microbiological approach to alleviate soil replant syndrome in peaches” written by Newberger et al., explained the replant (RS) on the growth and yield of tree fruit orchards. In addition, author also explained, that by developing/altering health bacteriome we can reduce the RS in peach. The study is well designed, introduction is good, methodology is sound, results are interested having potential of readers interest and manuscript is in the scope of journal. However, I have some suggestions which authors need to be addressed.

We thank you for taking the time to read the manuscript and we have addressed the following suggestions.

Q1; What is different between soil bacterial microbiome or soil bacteriome? In my opinion the words microbiome is a broad term dealing with all kinds of microorganism including bacteria, fungi, actinomycetes. In this study, authors just only study the soil bacteriome not the soil microbiome, so it’s better to be use a specific word “soil bacteriome” rather than using a broad word “soil bacterial microbiome”.

We agree that the term used should be soil bacteriome. This has been reflected in the manuscript.

Q2; What author means by cover crop growth? It is confusing use a proper terminology.

We used the term cover crop growth to refer to cover crop biomass or as a time point of when the cover crops were grown. We agree that this is not conventional terminology and have replaced the term with either biomass or stated this soil was from where the cover crops had been grown.

Q3; References are not in a proper order line 32 references start from 45, 50, 75, and 91. Where is reference 1? Same as for line34, 36, 38, and 39…………………………….

We misunderstood the reference guidelines and apologize. The numbering scheme we used was based on the citations in alphabetical order, not the first time it was referenced in the manuscript. The order of the references in text citations now reflects the style of the journal.

Q4; In the whole manuscript the scientific names are no italic, I do not know why authors are doing like this? I think authors cannot read the manuscript after following the journal format, as the references were also start form 45 not from 1. Authors should adopt a serious behavior when writing and formatting a manuscript.

We apologize for forgetting to italicize the scientific names. This issue was fixed throughout the manuscript.

Line 12; Prunus persica should be italic

The scientific name for peach is now in italics.

Line 13; followed by cover crop growth and incorporation the meanings are not clear need to be revise and same for line 14.

The meaning for cover crop growth and incorporation have been made more explicit.

Line 14-16; bacterial soil microbiome needs to be revise in the whole manuscript as “soil bacteriome”

We agree with this comment and proposed Q1 from before. We have also differentiated this term with “bulk soil bacteriome” and “rhizosphere bacteriome”.

Line 19; Scientific name must be italic and need to check in the whole manuscript.

We thank the reviewer for noticing this detail and the change has been made for the entire manuscript.

Line 36-38; I cannot understand what author want to say. Rewrite the sentence.

We agree that the sentence was not clear and have added details from the previous experiment and defined the “biotic component”.

Line 47; Can change diversity and composition of ……………

We have removed “disturbances can affect microbial growth and functionalities” and replaced it with “disturbances can affect bacteriome composition and functionality.” Additionally, it was verified that this is reflective of the citation. 

Line 72-75; rewrite the sentence

The sentence has been re-written and separated into two sentences.

Line 76; what do you mean by three 40-min liquid cycles?

For line 78, we mean that the soil was placed in autoclave bags and autoclaved on the 40-min liquid cycle at 121°C, which is a setting. In between cycles, bags carrying the soil were shaken so that the soil in the center of these bags was redistributed before being placed again in the autoclave for a second and third time. We have made it clearer that this was performed three times.

Line 78; for soil bacteriome analysis.

“Microbiome analysis” has been changed to “bacteriome analysis”.

Line 80-81; each pot was filled with 2.1 kg of soil either untreated………………………

Untreated and autoclaved soil description has been added.

Line 81-82; delete just below 5 cm of the lip of cup. Each pot contained c. 2.1 kg of soil.

We have deleted “just below 5 cm of the lip of cup” and added “2.1 kg of soil” to the previous line.

Line 89-92; Reference must be in number format.

References in the materials and methods have been changed into number format and follows the guideline of the journal Microorganisms.

Line 101; Everyone knows that how to separate a supernatant or liquid from the falcon tubes.

Seed sterilization description has been reduced.

Comments on the Quality of English Language

The sentence structure is not correct. incomplete sentences. Authors must revise the manuscript for sentence structure. Authors must delete unnecessary sentences, such as black plastic. Using a pipette, the NaOCl solution was then aspirated out.

Thank you for your suggestions. All suggestions were helpful and have been adapted as revisions for the manuscript. Additionally, we will review again to look for unnecessary sentences and send the manuscript to our writing center to improve on the sentence structure and to remedy the incomplete sentences.

Reviewer 2 Report

This is a very interesting study, with very interesting and notable results regarding RS and the microbiology of autoclaved and non-autoclaved, cover crop and no cover crop treatments and relationship to peach growth. The experiment is well-set-up and appropriate methodologies were used throughout. Although since fungal microorganisms are very important in the development and remediation of replant syndrome, the lack of the inclusion of any data regarding fungal microbiomes is disappointing and limits the usefulness of the results, the data on the soil bacterial microbiome still provides abundant, albeit incomplete, information. And overall, the paper is well-written. However, I do have some issues with the selected presentation and analysis of data, as well as some serious issues with the focus and interpretation of some of that data. 

I have included numerous comments and suggestions throughout the attached edited version of the manuscript that outlines these issues and other needed revisions.  Briefly, in addition to some minor issues, the main issues are the following: First, the soil analysis data are not presented at all (and soil respiration, although measured is not even mentioned), but should be, as is important in relation to other effects on microbiome. Most importantly, the focus of the microbiome data analyses presented as well as the interpretations made seem misguided, or at least do not emphasize the most important aspects or relationships. For example, emphasis on the increase of certain bacterial species in autoclaved soil vs. non-autoclaved soil in the cover crop trial, rather than the decrease or loss of species as being responsible for improved CC growth in autoclaved soil, as well as much analysis of the increase of bacterial species in non-autoclaved vs autoclaved soil for the peach growth trial, but no direct comparisons made for the non-autclaved CC soils vs. the non-autoclaved no cover soils, as this is where the critical difference in RS occurs, and where the true plant effects relevant to RS could be elucidate, yet are barely mentioned in this paper. Many further statements, especially regarding the importance of beneficial bacterial species such as B. megaterium in these interactions, are not supported by the data, as the highest abundance of B. megaterium was found in the raw RS soil, not in the remediated CC soils.  Such misleading interpretations led to some unwarranted conclusions regarding 'healthy' microbiomes, again referencing changes in populations of B.megaterium, when this relationship is NOT a good example, and does not seem to be related to the effects on RS due to CC inclusion. 

Overall, the authors need to re-think and re-evaluate the important interactions and comparisons to be made - what are the factors most related to the differences in RS observed in the peach trials, and how to best evaluate them. The authors have an excellent, very rich data set here, but need to revise how they approach, analyze, present, and interpret that data, because the way it is presented here is not the best presentation, and does not provide the most useful information, interpretations, or conclusions from that data. Thus, although the paper presents much excellent data and makes many important points, the way that data is analyzed and interpreted needs to be thoroughly revised to provide more useful conclusions that are better supported by the data.  As mentioned, more thorough comments and suggestions are provided in the attached edited document

Overall good, some minor misspellings or sentence structure issues, some redundant passages. Just read through carefully prior to resubmission.

Author Response

Dear reviewer 2,

Please find our point-by-point response to the comments of the reviewers.

Sincerely,

First Author

Derek R. Newberger, PhD Candidate

Graduate research assistant, Department of Horticulture and Landscape Architecture

Colorado State University

Derek.Newberger@colostate.edu

Dr. Jorge M. Vivanco, PhD

Professor, Department of Horticulture and Landscape Architecture

Colorado State University

J.Vivanco@colostate.edu

Reviewer 2:

This is a very interesting study, with very interesting and notable results regarding RS and the microbiology of autoclaved and non-autoclaved, cover crop and no cover crop treatments and relationship to peach growth. The experiment is well-set-up and appropriate methodologies were used throughout. Although since fungal microorganisms are very important in the development and remediation of replant syndrome, the lack of the inclusion of any data regarding fungal microbiomes is disappointing and limits the usefulness of the results, the data on the soil bacterial microbiome still provides abundant, albeit incomplete, information.

We agree that the fungal community is critical to understanding how RS develops and persists. However, the goal of this study was to focus on sustainable agricultural techniques which promoted peach health and identify potential plant growth promoting rhizobacteria (PGPRs).

And overall, the paper is well-written. However, I do have some issues with the selected presentation and analysis of data, as well as some serious issues with the focus and interpretation of some of that data. I have included numerous comments and suggestions throughout the attached edited version of the manuscript that outlines these issues and other needed revisions.

We thank the reviewer for their comments and suggestions throughout the manuscript. Most of the recommended changes were made.

Briefly, in addition to some minor issues, the main issues are the following: First, the soil analysis data are not presented at all (and soil respiration, although measured is not even mentioned), but should be, as is important in relation to other effects on microbiome.

We agree that the soil analysis data should be shown in a more prominent fashion. We have made a table which shows the soil nutrients by autoclaved and non-autoclaved soils and placed it in the supplementary section.

Most importantly, the focus of the microbiome data analyses presented as well as the interpretations made seem misguided, or at least do not emphasize the most important aspects or relationships. For example, emphasis on the increase of certain bacterial species in autoclaved soil vs. non-autoclaved soil in the cover crop trial, rather than the decrease or loss of species as being responsible for improved CC growth in autoclaved soil, as well as much analysis of the increase of bacterial species in non-autoclaved vs autoclaved soil for the peach growth trial, but no direct comparisons made for the non-autoclaved CC soils vs. the non-autoclaved no cover soils, as this is where the critical difference in RS occurs, and where the true plant effects relevant to RS could be elucidate, yet are barely mentioned in this paper.

We agree that a bacterial species comparison with the control, non-autoclaved no cover soils, and the successful treatments, non-autoclaved CC soils, should be made to highlight the effect of the cover crop treatment. However, we have focused our analyses on PGPRs because plenty of studies have described the presence of soil pathogens as a possible cause of RS.

Many further statements, especially regarding the importance of beneficial bacterial species such as B. megaterium in these interactions, are not supported by the data, as the highest abundance of B. megaterium was found in the raw RS soil, not in the remediated CC soils. Such misleading interpretations led to some unwarranted conclusions regarding 'healthy' microbiomes, again referencing changes in populations of B.megaterium, when this relationship is NOT a good example, and does not seem to be related to the effects on RS due to CC inclusion.

We agree with the comments of the reviewer. Our intention was not to uniquely highlight B. megaterium, more than Gaiella occulta, and Nitrospira japonica, and especially not more so than Paenibacillus castaneae and Bellilinea caldifistulae which were found to exclusively colonize the peach rhizosphere in non-autoclaved cover crop treatments. Our goal is to use B. megaterium to highlight the consequences of soil disinfection.

We understand how this issue is related to Supplementary Figure 4 in which the “RS Control” (initial soil collected before the experiment began) was found to have the highest abundance of B. megaterium compared to all other non-autoclaved soils, including the no crop control, cover crop bulk soils, peach bulk soil, and peach rhizosphere. However, we would like to point out that the comparison made should use the abundance of B. megaterium as noted by the light grey box over the “Rotation Crop” to following cropping cycle treatments.

The intention of showing the “RS Control” was to highlight that B. megaterium was present in the original soil and was not coming from another source (greenhouse, tap water, or researcher).

Although it is well established that B. megaterium is a beneficial bacterium, the role it may play specifically replant syndrome is unknown. Our analysis compared the presence of B. megaterium in the rhizosphere of peaches grown in autoclaved soils to the rhizosphere of peaches grown in non-autoclaved soils. In section 4.2, we emphasize that we aimed to “highlight the loss of key beneficial bacterial species which did not recover their abundancies post autoclaving, Bacillus megaterium is used as an example”. Additionally, cover crop treatments, which did show some promise to alleviate RS, did not prevent the recruitment of a beneficial such as B. megaterium. At no point in the paper do we say that B. megaterium was the main factor involved in peach biomass alleviation due to RS. In the conclusion we state that “non-autoclaved soils which had higher peach biomass and most likely relatively ‘healthier’ microbiomes consisted of an increased abundance of Bacillus megaterium, Gaiella occulta, and Nitrospira japonica among other potential beneficial microbes. These bacteria could be specifically recruited by peaches since abundances where present in the peach rhizosphere in all non-autoclaved treatments”. Regardless, we agree that further study should be made to infer the role B. megaterium plays in alleviating RS. We have added a sentence that explicitly mentions that B. megaterium and other bacterial taxa were also present in the no cover crop control which did not outperform the non-autoclaved cover crop treatments (alfalfa, corn, and tomato) in terms of biomass. We have reviewed the sections mentioning B. megaterium and reworded the sentence in the conclusion to ensure we avoid making strong correlations between abundance and biomass when we intend to highlight how soil disinfection techniques can remove beneficial bacteria which failed to recover for the duration of the study.

Overall, the authors need to re-think and re-evaluate the important interactions and comparisons to be made - what are the factors most related to the differences in RS observed in the peach trials, and how to best evaluate them.

We agree and thank the reviewer for the idea to make a bacterial species comparison with the unsuccessful treatment of non-autoclaved and no cover crop control, and the successful treatments of non-autoclaved cover crops. The manuscript highlights this difference in biomass, and by highlighting the importance of beneficial bacterial species were cultivated exclusively in the rhizosphere of non-autoclaved soils by peaches such and Peribacillus simplex and Arenimicrobium luteum for all cover crop histories would be an improvement.

The authors have an excellent, very rich data set here, but need to revise how they approach, analyze, present, and interpret that data, because the way it is presented here is not the best presentation, and does not provide the most useful information, interpretations, or conclusions from that data. Thus, although the paper presents much excellent data and makes many important points, the way that data is analyzed and interpreted needs to be thoroughly revised to provide more useful conclusions that are better supported by the data. As mentioned, more thorough comments and suggestions are provided in the attached edited documentpeer-review-29086274.v2.pdf

We thank the reviewer for taking the time to make edits to the manuscript itself, and for adding ideas on how to include a deeper interpretation of the data.

Comments on the Quality of English Language

Overall good, some minor misspellings or sentence structure issues, some redundant passages. Just read through carefully prior to resubmission. (/user/review/displayFile/38610164/yMjsi21G? file=review&report=29086274)

Reviewer 3 Report

I have carefully reviewed the manuscript entitled "A microbiological approach to alleviate soil replant syndrome in peaches" by Newberger et al. submitted to the journal Microorganisms. Overall, This is a good study that yields many useful data benefitting to broaden the understanding of peach replant syndrome by providing an insight into soil microbial ecology. In addition, the third-generation sequencing technology used here (nanopore sequencing) helps to detect the key bacterial taxa capable of controling RS problem at a finer microbial taxonomy level, such as Bacillus megaterium, Gaiella occulta, and Nitrospira japonica. The explanation for observed results by authors is factual and reasonable. I just have some minor problems as the following:

1. Why was fungal community neglected in the current study?

2. Whether there were significant differences in soil total bacterial abundance between treatments? 

3. The contents and descriptions involved in the alpha diversity of bacterial community should be supplemented in the manuscript main text, rather than just Figure S6.

4. Whether there exist direct significant linear/non-linear correlations between beneficial bacterial taxa detected here and peach biomass/cover crops biomass? 

Author Response

Dear reviewer 3,

Please find our point-by-point response to the comments of the reviewers.

Sincerely,

First Author

Derek R. Newberger, PhD Candidate

Graduate research assistant, Department of Horticulture and Landscape Architecture

Colorado State University

Derek.Newberger@colostate.edu

Dr. Jorge M. Vivanco, PhD

Professor, Department of Horticulture and Landscape Architecture

Colorado State University

J.Vivanco@colostate.edu

Reviewer 3:

I have carefully reviewed the manuscript entitled "A microbiological approach to alleviate soil replant syndrome in peaches "by Newberger et al. submitted to the journal Microorganisms. Overall, this is a good study that yields many useful data benefitting to broaden the understanding of peach replant syndrome by providing an insight into soil microbial ecology. In addition, the third-generation sequencing technology used here (nanopore sequencing) helps to detect the key bacterial taxa capable of controlling RS problem at a finer microbial taxonomy level, such as Bacillus megaterium, Gaiella occulta, and Nitrospira japonica. The explanation for observed results by authors is factual and reasonable.

We thank the reviewer for agreeing that the data shows promise to provide insights to RS soils. We propose that soil disinfection can cause the loss of beneficial bacteria, with further investigation of the role Bacillus megaterium, Gaiella occulta, and Nitrospira japonica may play in alleviating RS.

I just have some minor problems as the following:

  1. Why was fungal community neglected in the current study?

We agree that the fungal community is critical to understanding how RS develops and persists. There have been several studies identifying repeated key fungal phytopathogens and new potential phytopathogens. The same has been done for bacteria, oomycetes, and nematodes. However, the goal of this study was to focus on sustainable agricultural techniques which promoted peach health and identify potential plant growth promoting rhizobacteria (PGPRs). We focused on PGPRs since current tools for inoculation of single or mock communities of bacteria have been more developed than those for beneficial fungi.

  1. Whether there were significant differences in soil total bacterial abundance between treatments?

We thank the reviewer for this suggestion. We did not report our qPCR data in the manuscript and believe will instead, discuss the results of the Shannon index in the manuscript.

  1. The contents and descriptions involved in the alpha diversity of bacterial community should be supplemented in the manuscript main text, rather than just Figure S6.

We agree that the alpha diversity of the bacterial communities should be referenced in the main text. A new figure was made to remove controls and to visualize the data in boxplots to better display the data. We have incorporated text about it in the Results section. However, we will not have a figure in the main text since this figure shows only trends. The figure will remain as Supplementary Figure 6.

  1. Whether there exist direct significant linear/non-linear correlations between beneficial bacterial taxa detected here and peach biomass/cover crops biomass?

We agree that having a figure to correlate beneficial bacterial taxa with peach/cover crop biomass would be important for the purpose of this manuscript, however, linear/non-linear correlations would not support this. Supplementary Figure 4 shows Bacillus megaterium relative abundance in the bulk soil/rhizosphere by cover crop treatment and no crop control. This figure indicates that peaches with the highest biomass, also had a high relative abundance of Bacillus megaterium. However, B. megaterium. was also present in non-autoclaved soils with no cover crop treatment, and peach biomass for this treatment was not higher than soils without B. megaterium. More studies would be required to see if B. megaterium could directly reduce RS for peaches.  The aim of the present manuscript was to find agricultural practices to increase beneficial bacteria to reduce RS, and that soil disinfection could remove such beneficial bacteria.

Round 2

Reviewer 1 Report

Dear Editor 

Authors respond the queries very well.

The quality of manuscript is improved.

Good luck 

Author Response

Dear reviewer 1,

Please find our point-by-point response to the comments for round 2 of the comments.

Sincerely,

First Author

Derek R. Newberger, PhD Candidate

Graduate research assistant, Department of Horticulture and Landscape Architecture

Colorado State University

Derek.Newberger@colostate.edu

Dr. Jorge M. Vivanco, PhD

Professor, Department of Horticulture and Landscape Architecture

Colorado State University

J.Vivanco@colostate.edu

Reviewer 1:

 Authors respond the queries very well. The quality of manuscript is improved. Good luck

We thank the reviewer for their comments, for we believe their contribution improved the manuscript.

Reviewer 2 Report

The authors have provided detailed responses to the previous comments and suggestions and have made numerous improvements, addressing the most serious issues, in this revised version. However, some of the issues have only been partially addressed, and although the authors seemed to acknowledge and agree with some comments, only minor modifications or adjustments have been made to the manuscript itself. Thus, although the revised manuscript is much improved, there are still some issues that need more attention prior to publication. 

For example, The authors have added a supplementary table showing soil properties among the treatments. However, this table does not include soil respiration results, which are still not mentioned anywhere in the paper. Of all the measured soil properties, soil respiration is the most critical for this study, as it shows the overall microbial activity (rather than individual bacterial abundance), which may be very important comparison among the soil treatments. If measured, I do not understand why these data would not be presented and discussed, as it is very appropriate in relation to the other bacteriome data. 

Also, the authors present in their response some justification for not including analysis of the fungal microbiome in these studies, however they did not include this info in the manuscript itself. This will be a question asked by virtually all readers and your justification for only assessing the bacteriome needs to be stated right in the Introduction, as it relates to the objectives of the research. 

Regarding my main issue with the previous version of the paper, the need to re-focus attention on differences that may be related to RS and remediation of RS, the authors have made some adjustments so that the interpretations and conclusions are more consistent with the data. However, they have not added any further info or discussion regarding key differences between the non-autoclaved no CC soil and the various non-autoclaved CC treated soils, because that is where the key difference in RS symptoms occur. For example, the authors do now make specific reference to the 2 bacteria species that increased in the CC soils, but not the NCC soils, which is an improvement, but there are several other bacterial species (8) that were also increased in multiple CC soils (2 or more) and not in NCC soils, which could also play some role and should be mentioned. These may be more important to note than the long list of species presented that were observed to uniquely increase with each CC, since these were increased with multiple different CCs. I still feel there is too much emphasis placed on the auto vs non-auto, and not enough on these differences within non-auto soils with and without CC, but this is your paper, and I am now OK with whatever interpretations you want to make, as long as they are supported by the data.

There are some other minor corrections needed. For example, I'm not sure why all the figures and tables were moved to the end of the Results instead of within the section near where they are referenced, but they should be moved back to near the first mention in text. 

In Figure 4. Differences letter designations start with ab, rather than a. Where is a? There has to be an a if there is an ab. If no a, then need to re-letter the designations appropriately.

Revised sentence at Line 561 doesn't make sense as written, as it says 'Although the non-autoclaved treatment had the lowest peach biomass,' which does not correspond with the data. What non-autoclaved treatment? Generally, the non-autoclaved  treatments had the highest peach biomass, so not clear what is being said here.

English quality good, may be a few minor typos/corrections needed

Author Response

Dear reviewer 2,

Please find our point-by-point response to the comments for round 2 of the comments.

Sincerely,

First Author

Derek R. Newberger, PhD Candidate

Graduate research assistant, Department of Horticulture and Landscape Architecture

Colorado State University

Derek.Newberger@colostate.edu

Dr. Jorge M. Vivanco, PhD

Professor, Department of Horticulture and Landscape Architecture

Colorado State University

J.Vivanco@colostate.edu

Reviewer 2:

The authors have provided detailed responses to the previous comments and suggestions and have made numerous improvements, addressing the most serious issues, in this revised version. However, some of the issues have only been partially addressed, and although the authors seemed to acknowledge and agree with some comments, only minor modifications or adjustments have been made to the manuscript itself. Thus, although the revised manuscript is much improved, there are still some issues that need more attention prior to publication.

We thank the reviewer for taking the time to make comments within the manuscript itself, and they were constructive and specific. We have been more attentive on the persisting issues within the manuscript. We aimed to fully address the issue of the soil respiration results, the justification not including analysis of the fungal microbiome, the minor revisions to figure letter designations, and Line 561. We partially addressed the recommended revisions related to the differential abundance analysis since we highlighted bacteria which were unique for two or more cover crops but continue to emphasize the difference between autoclaved and non-autoclaved soils.

For example, the authors have added a supplementary table showing soil properties among the treatments. However, this table does not include soil respiration results, which are still not mentioned anywhere in the paper. Of all the measured soil properties, soil respiration is the most critical for this study, as it shows the overall microbial activity (rather than individual bacterial abundance), which may be very important comparison among the soil treatments. If measured, I do not understand why these data would not be presented and discussed, as it is very appropriate in relation to the other bacteriome data.

Regarding Table S1, the table now keeps the order of alfalfa, fescue, corn, tomato, and none as the rest of the manuscript but has the autoclaved and non-autoclaved separated as readers are probably more interested in seeing the impact autoclaving has on the soil nutrients. Furthermore, we did measure soil respiration and have added this data to Table S1. We thank the reviewer for this overlooked parameter. Here is a figure which is in neither the manuscript or the supplementary and is for the reviewer’s view only. Although a respiration figure loosely matches the biomass trends for alfalfa, fescue, and corn, the tomato and no-crop control are inconsistent, and we do not believe that it would be beneficial to be discussed in the manuscript itself.

(this figure is visible in the uploaded document)

Also, the authors present in their response some justification for not including analysis of the fungal microbiome in these studies, however they did not include this info in the manuscript itself. This will be a question asked by virtually all readers and your justification for only assessing the bacteriome needs to be stated right in the Introduction, as it relates to the objectives of the research.

We have included justification for not including analysis of the fungal microbiome at the end of the introduction. We agree that many readers will be looking to see the bacteriome by complimented by not only the mycobiome, but oomycete and nematode abundancies as well. We have emphasized that the scope of this study is to identify an agricultural practice which increases known and possibly new plant growth promoting bacteria, and that possible phytopathogenic culprits (bacterial, fungal, oomycetes, and nematodes) have been the focus of previous articles.

Regarding my main issue with the previous version of the paper, the need to re-focus attention on differences that may be related to RS and remediation of RS, the authors have made some adjustments so that the interpretations and conclusions are more consistent with the data. However, they have not added any further info or discussion regarding key differences between the non-autoclaved no CC soil and the various non-autoclaved CC treated soils, because that is where the key difference in RS symptoms occur. For example, the authors do now make specific reference to the 2 bacteria species that increased in the CC soils, but not the NCC soils, which is an improvement, but there are several other bacterial species (8) that were also increased in multiple CC soils (2 or more) and not in NCC soils, which could also play some role and should be mentioned. These may be more important to note than the long list of species presented that were observed to uniquely increase with each CC, since these were increased with multiple different CCs. I still feel there is too much emphasis placed on the auto vs non-auto, and not enough on these differences within non-auto soils with and without CC, but this is your paper, and I am now OK with whatever interpretations you want to make, as long as they are supported by the data.

We agree with the reviewer that the key differences between the non-autoclaved no CC soil and the various non-autoclaved CC treated soils is an important comparison since the microbiomes are the most similar yet there is still the biomass reduction. For this reason, it would have been ideal to highlight bacteria which increased in all three of the cover crops (alfalfa, corn, and tomato) to reflect the biomass figure, but as noticed, there are none for just those three and is the reason we decided to highlight unique bacteria per cover crop. We see the validity in highlighting other bacterial species that were also increased in 2-3 CC soils and not in NCC soils, and have added this into the manuscript. We understand that there needs to be an emphasis on the consortia of bacteria which reduce RS, and that would not be best represented by focusing on the unique bacteria by individual cover crop. Nonetheless, we aimed to show how each cover crop treatment brought their own unique set of beneficial bacteria and that there may be multiple different bacteriomes that can alleviate RS, we just do not know yet which microbes play a role. We have highlighted why we believe that it is important to mention these “one-off” bacteria as different strategies to reduce RS.

For the intent to provide a solution to RS, the most important is looking at the non-auto soils with and without CC, however, the use of cover crops to alleviate RS is known to have varying degrees of success. We believed that to help explain the shift from autoclaving being beneficial for cover crop growth but not in the second crop cycle using peaches needed a fair amount attention since it was unexpected.

Lastly, as mentioned in the first round, this manuscript does contain a lot of data and there are many perspectives one could take for analysis and interpretation, and we value your comments and redirection to use the non-autoclaved no CC soil in our bacteriome analysis.

There are some other minor corrections needed. For example, I'm not sure why all the figures and tables were moved to the end of the Results instead of within the section near where they are referenced, but they should be moved back to near the first mention in text.

We did not move the figures to the end of results. We believe that the journal did as was one of the formatting changes indicated by this statement which is above the download manuscript button “Please download the latest version of the manuscript for revision. Your original submission may have been changed” and we are unaware if their current placement will be their final placement.  

In Figure 4. Differences letter designations start with ab, rather than a. Where is a? There has to be an a if there is an ab. If no a, then need to re-letter the designations appropriately.

We thank the reviewer for this recommendation and have modified the figure to start with the letter designation of “a”. We have verified the re-lettering of the designations with the text output of the pairwise comparison from RStudio.

Revised sentence at Line 561 doesn't make sense as written, as it says 'Although the non-autoclaved treatment had the lowest peach biomass,' which does not correspond with the data. What non-autoclaved treatment? Generally, the non-autoclaved treatments had the highest peach biomass, so not clear what is being said here.

We agree with the reviewer, and the non-autoclaved treatment we were referring to the control (non-autoclaved soil with no history of cover crops) which had the lowest biomass of all the non-autoclaved treatments. Line 561 has been revised to “Although the control (non-autoclaved soil with no history of cover crops) had the lowest peach biomass out of the non-autoclaved treatments, the control showed the highest count of unique bacterial taxa (n = 53), many of which have been previously associated with beneficial traits.”
